# Ducted wind turbines in yawed flow: A numerical study

Vinit Dighe[1], Dhruv Suri[2], Francesco Avallone[1], and Gerard van Bussel[1]

[1]Wind Energy Research Group, Faculty of Aerospace Engineering, Technological University of Delft, Netherlands
[2]Renewable Energy Research Group, Department of Aeronautical and Automobile Engineering, Manipal Institute of Technology, India

**Correspondence:** Vinit Dighe (v.v.dighe@tudelft.nl)

**Abstract.** Ducted Wind Turbines (DWTs) can be used for energy harvesting in urban areas where non-uniform flows are caused by the presence of buildings or other surface discontinuities. For this reason, the aerodynamic performance of DWTs in yawed flow conditions must be characterized depending upon its geometric parameters and operating conditions. A numerical study to investigate the characteristics of flow around two DWT configurations using a simplified duct-actuator disc (AD) model is carried out. The analysis shows that the aerodynamic performance of a DWT in yawed flow is dependent on the mutual interactions between the duct and the AD; an interaction that changes with duct geometry. For the two configurations studied, the high cambered variant duct configuration returns a gain in performance by approximately 11% up to a specific yaw angle ($\alpha = 17.5°$) when compared to the non-yawed case; thereafter any further increase of yaw angle results in a performance drop. On the contrary, performance of less cambered variant duct configuration drops for $\alpha > 0°$. The gain in the aerodynamic performance is attributed to the additional camber of the duct that acts as a flow conditioning device and delays duct wall flow separation inside of the duct for a broad range of yaw angles.

## 1 Introduction

Global energy demand is expected to more than double by 2050 owing to the growth in population and economy (Gielen et al., 2019). The global wind power capacity quadrupled in less than a decade reaching 597 Gigawatt by the end of 2018 compared to 120 Gigawatt in 2008 (Dupont et al., 2018). Wind turbines are typically installed away from populated areas. This necessitates the transfer of electricity via grids over large distances, which increases the levelized cost of electricity (LCOE). Integration of wind turbines into urban areas is challenging; the presence of buildings, trees and surface discontinuities lead to lower wind speed, non-uniform inflow and larger turbulent fluctuations compared to open fields. The key parameters identified in the turbine design space are those relating to performance and those relating to cost (Valyou and Visser, 2020). To address the performance-related challenges, design modifications of wind turbines, suitable for operation in an urban setting is required.

A possible technological solution to extract wind energy in urban areas is represented by Ducted Wind Turbines (DWTs). DWTs increase energy extraction with respect to conventional horizontal axis wind turbines (HAWTs) for a given turbine radius and free-stream velocity (van Bussel, 2007). DWTs are constituted of a turbine and a duct (also named as diffuser or shroud); the role of the latter is to increase the flow rate through the turbine relative to a similar turbine operating in the open atmosphere, thus increasing the generated power. Its aerodynamic working principle is best explained as the generation of a

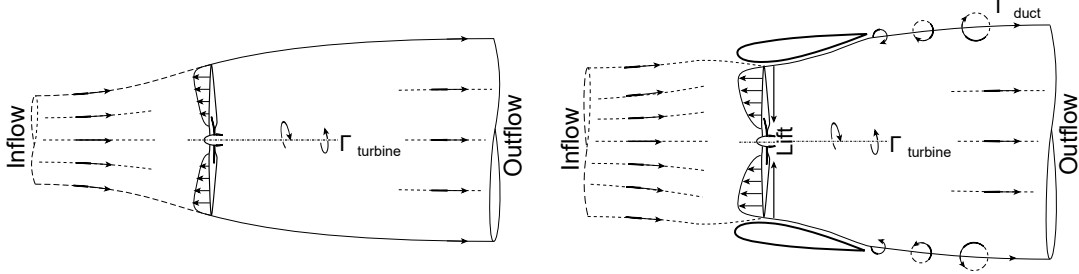

**Figure 1.** Schematic of stream-tube model for a bare turbine (left) and DWT (right). The trailing vorticity in the wake is denoted by $\Gamma$.

radial force upon the flow. A force towards the DWTs center-line will cause an expansion of flow downstream of the turbine beyond what is attainable for a bare wind turbine. This provides a reduced pressure behind the turbine, and hence an increased mass flow through the turbine (van Bussel, 2007). For an aerodynamically shaped duct, the sectional lift force of the duct is directed inboard, but this lift will be tilted slightly in upwind direction when an axial force on the turbine is present. The associated bound vorticity (see Figure 1) on the duct induces the increased the mass flow through the turbine (de Vries, 1979). A significant amount of literature on DWTs, based on the combined use of theoretical, numerical and experimental techniques, exists (Igra, 1981; Gilbert and Foreman, 1983; Abe et al., 2005; Toshimitsu et al., 2008; Werle and Presz, 2008; Khamlaj and Rumpfkeil, 2017). Questions over the performance of DWTs in yawed flow remain, however.

Igra (1981) studied experimentally the effects of yaw on the performance of DWTs. Eight geometries were investigated using different duct profiles and an actuator disc (AD) model to represent the turbine. The eight configurations differed in the duct expansion ratio, i.e., the ratio of exit area of the duct to the turbine area. The AD with a thrust coefficient of approximately 0.5 was chosen. It was found that when the duct expansion ratio was less than 4.5, little or no difference in the power output was measured up to a yaw angle of $\pm$ 30° while any further increase in yaw resulted in power reduction. On the other hand, when the duct expansion ratio was higher than 4.5, the generated power decreased even for small yaw angles. Igra explained that the yaw insensitivity, for the low duct expansion ratio configurations, is due to the lift force increase by the annular duct section. The author did not provide any explanation to further clarify the physics behind performance drop for large duct expansion ratio. On the same line, researchers from Grumman Aerospace tested a bare turbine and two DWT models (named as Baseline DAWT and DAWT 45) varying the yaw angle up to 40° with increments of 10° (Gilbert and Foreman, 1983). Both the Baseline DAWT and DAWT 45 models showed a negligible change in the power up to a yaw angle of 30°, and a drastic reduction in power at yaw angle of 40°. Surprisingly, the bare turbine also demonstrated no dependence on the yaw angle up to 30°. They stated that this was due to the long center-body configuration, similar in all the three designs, that helped channeling the incoming flow towards the upwind turbine blade and at the same time shielding the downwind turbine blade, thus offering an insensitivity to yaw. However, in a follow up paper (Foreman and Gilbert, 1983) they stated that these yaw tests were inconclusive whether the yaw insensitivity was due to the center-body effect or the duct geometry itself. More recently, Phillips et al. (2002) combined experimental and numerical analysis to study DWTs under yawed flow. They concluded that the power increase for a DWT in yawed flow can only be achieved with a slotted duct design (named as Mo), with the added mass flow

of air through the slot increasing the boundary layer flow control and preventing flow separation over the suction side (inner surface) of the duct under severe yaw misalignment. The above literature, due to the contrasting nature of the conclusions, lacks clarity on the aerodynamics of DWTs in yawed flow, and particularly on the effect of the duct geometry on the aerodynamic performances. The present article aims to reignite the insights of Igra (1981), Foreman and Gilbert (1983) and Phillips et al. (2002) to study the effects of yaw on the performance of DWTs based on a numerical study.

In all the simulations presented in this article, the turbine is represented using a numerical actuator disc (AD) model, a method widely used to model the principal effects of turbine in a simplified manner. In the AD model, the turbine forces are assumed to be distributed evenly along the AD; hence, the influence of the blades is taken as an integrated quantity in the azimuthal direction. The effects of distributed forces for real turbine geometries are modelled using more sophisticated techniques like actuator line (Troldborg, 2009) or actuator surface (Shen et al., 2009) methods. Incorporating the real turbine geometries, which would necessarily have to be different for ducted and for bare operation, would confuse turbine and duct effects, preventing a proper analysis of DWTs in yawed flow. Thus, the AD approach is chosen deliberately for this investigation, so as to study the impact of duct shapes, and not the specific performance of a rotor within a duct. The effects of real turbine within different duct geometries are studied in a subsequent publication by the authors, see Dighe et al. (2020). The numerical AD method has been extensively validated, see for example Dighe et al. (2019a, b). The numerical AD model has been been applied by Mikkelsen and Sørensen (2001) to study the flow on a horizontal axis wind turbine in axial and yawed flow conditions. The numerical predictions agree reasonably well, both in axial and yawed flow conditions, when compared to the measurements on the Tjæreborg 2MW field turbine. This model is also employed by Tongchitpakdee et al. (2005) to study yaw; the NASA-Ames experiments of the NREL Phase VI turbine are modeled for yaw angles from $0°$ to $45°$ to find reasonable agreement with the experiments.

The paper is organized as follows. Section 2 reports the non-dimensional coefficients adopted for characterizing the aerodynamic performance of the duct-AD model, both under non-yawed and yawed flow conditions. Section 3 describes the numerical settings and parameters with the description of the duct profiles chosen for the current investigation. Section 4 reports the numerical validation study. Insights on the aerodynamic performance coefficients with respect to yawed flow will be discussed in section 5, together with flow analysis. Finally, the most relevant results are summarized in the conclusions.

## 2   Duct - AD flow model

The turbine is modelled by a flat AD. The AD exerts a constant thrust force $T_{AD}$, calculated across the AD surface $S_{AD}$, which corresponds to a non-dimensional thrust force coefficient:

$$C_{T_{AD}} = \frac{T_{AD}}{0.5\rho U_\infty{}^2 S_{AD}}, \tag{1}$$

where $\rho$ is the fluid density and $U_\infty$ is the free-stream velocity.

To generate $T_{AD}$, a uniform pressure drop is present across the AD surface, $T_{AD} = \Delta p \times S_{AD}$. The pressure drop $\Delta p$ is taken from experiments (Tang et al., 2016) and is given as an input parameter to the numerical simulations. The mean velocity

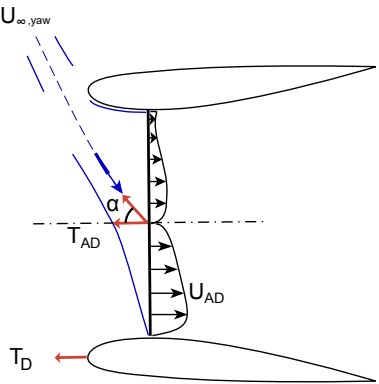

**Figure 2.** Schematic of yawed flow around a duct-AD model

across the AD radial plane, which is a function of AD thrust coefficient $U_{AD_0} = f(C_{T,AD})$, can be expressed by integrating the difference of the free-stream velocity component $U_x$ across the AD surface:

$$\frac{U_{AD_0}}{U_\infty} = \frac{1}{S_{AD}} \oint_{S_{AD}} \frac{U_x}{U_\infty} dS. \tag{2}$$

Using Eqs. 1 and 2, the power coefficient for a bare AD reads:

$$C_{P_o} = \frac{U_{AD_o}}{U_\infty} C_{T_{AD}}. \tag{3}$$

The subscript $o$ has been adopted for quantities evaluated for bare AD configuration.

For a duct-AD configuration, an additional thrust force exerted by the duct on the flow, or vice-versa, appears. Then, the total thrust force $T$ is the vectorial sum of the AD thrust force $T_{AD}$, and the duct thrust force $T_D$, given by:

$$T = T_{AD} + T_D. \tag{4}$$

The total thrust coefficient is then defined as:

$$C_T = C_{T_{AD}} + C_{T_D}. \tag{5}$$

Note that the duct thrust coefficient $C_{T_D}$ is normalized with the AD area $S_{AD}$ to facilitate direct addition to the AD thrust coefficient $C_{T_{AD}}$ for calculating the total thrust coefficient $C_T$. Then, the mean velocity at the AD for a duct-AD model is a bivariate function of AD thrust coefficient and the duct thrust coefficient: $U_{AD} = f(C_{T_{AD}} + C_{T_D}) = f(C_T)$. Similar to Eq. 3, the power coefficient for the duct-AD model, using $S_{AD}$ as the reference area, becomes:

$$C_P = \frac{U_{AD}}{U_\infty} C_T. \tag{6}$$

The power coefficient expression in Eq. 6 challenges the well-known Lanchester–Betz–Joukowsky limit of $\frac{16}{27}$ for maximum power coefficient obtainable for a HAWT. This should not appear like a surprising result, since the mass flow for a given $C_{T_{AD}}$

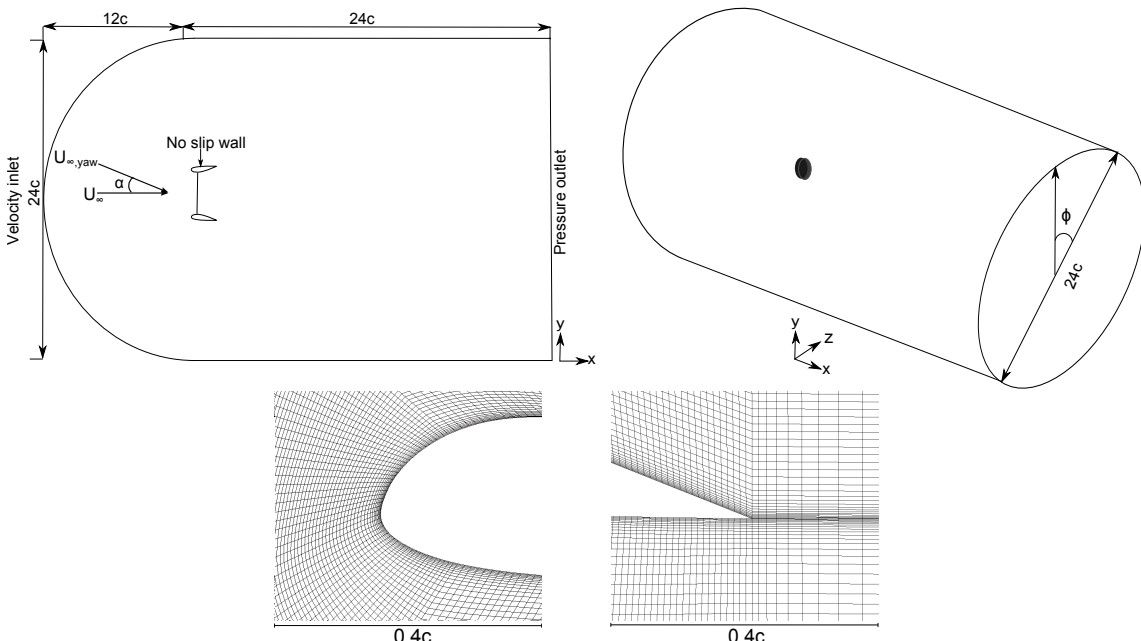

**Figure 3.** Computational domain showing the boundary conditions employed (top). The lengths are normalized with the duct chord length $c$. Representative, not to scale. Computational grid surrounding the leading and trailing edge of the duct (bottom).

is larger than the mass flow without a duct. The additional thrust needed for the momentum balance is offered by the tilting of the lift force on the duct in the direction towards the incoming wind. The above relations are also valid for a DWT under yawed flow condition. Figure 2 shows the schematic of flow around the duct-AD model, where $\alpha$ is the yaw angle relative to the incident free-stream direction.

## 3   Methodology and Computational Setup

In this study, a commercial CFD solver ANSYS Fluent® is employed for solving the governing flow equations. The more sophisticated Large Eddy Simulation (LES) method, used in the context of DWT modelling (Dighe et al., 2020), are more likely to be more accurate in resolving complex flow features such as flow separation and vortex shedding. However, LES method remains challenging for the parametric study presented here due to the limited computational capacity. Large flow separation regions are expected for DWTs in yawed flow. Flow solutions obtained using steady RANS formulation for DWTs with large yaw angles did not converge or even diverge. Moreover, the results presented by Phillips et al. (2008) show that the power predicted by the CFD simulations was significantly higher than that reached in the wind tunnel experiment. The over-prediction can be attributed to a very high blockage correction factor used, while for the the CFD results, the discrepancy can be attributed to the flow separation occurring inside the duct that was not captured computationally through the use of steady-state simulations and the choice of turbulence model (k-$\epsilon$). Therefore, the solver utilizes the unsteady Reynolds-averaged

Navier-Stokes (URANS) formulation to capture the asymptotic behavior (quasi-steady state) of the flow. The k-$\omega$ shear stress transport (SST) model is employed for the turbulence closure scheme. Apsley and Leschziner (2000) investigated the ability of various second-order closure models to predict separated flows in a duct and compared them to experimental data. $k - \omega$ SST model returns better predictions than the other second-order closure models with regards to approximating the unsteady flow in the velocity profiles of the duct. Moreover, Shives and Crawford (2012) investigated the application of different closure models for modelling ducted turbine flows. It was concluded that $k - \omega$ SST model outperforms the other first and second-order closure models. A pressure-based coupled solver was selected with a second order implicit transient formulation for improved accuracy. All solution variables were solved via second order upwind discretisation scheme.

In order to evaluate the numerical duct-AD model in nearly unconstrained flow, the computational domain extends $12c$ upstream and $24c$ downstream, where $c$ is the duct chord length. The distances are found to be safe choices to minimize the effects of blockage and uncertainty in the boundary conditions on the results; please refer to Appendix A. Using the finite volume method, the computational domain is discretized spatially into finite number of small control volumes known as grid. The grid have been generated using the commercial software ANSYS ICEM CFD. For the present computations, a C-grid structured zonal approach is chosen, see Figure 3, which proved advantageous in the case of a curved boundary, i.e. duct's leading edge. The C-shaped loop terminates in the wake region. The computational grid consists of quadrilateral cells with maximum $y^+$ value of $\approx 1$ on the duct wall. A 3D grid is created by extruding the 2D grid using 100 grid points in azimuthal direction $\phi$ using the surface grid extrusion technique (ANSYS, 2018). Boundary conditions are: uniform velocity at the inlet, zero gauge static pressure at the outlet, no-slip walls for duct surfaces. The numerical study is performed at a fixed $Re$ of $4.5 \times 10^5$. The influence of AD is included into the domain as an additional body force acting opposite to the direction of flow. This is achieved using a reverse fan boundary condition in ANSYS Fluent®. For a uniform thrust loading, the thrust force is given by:

$$T_{AD} = 0.5 C_{T_{AD}} \rho U_\infty^2, \tag{7}$$

where $C_{T_{AD}}$ is calculated from a semi-empirical relation of pressure drop curve and the velocity at the AD obtained from wind tunnel experiments. The fluid is air with fluid density $\rho = 1.276 \frac{kg}{m^3}$ and dynamic viscosity $\mu = 1.722 \times 10^{-5}$ $Pa \cdot s$. Values of free-stream velocity $U_\infty$ and turbulence intensity $I$ are chosen for consistency with the wind tunnel experiments. To establish yawed inflow conditions, the flow is rotated around the center-line axis by yaw angle $\alpha$ for different test cases.

The simulations were advanced through time with a CFL (Courant) number of one, which resulted in a time-step of approximately $2.67 \times 10^{-4}$ s. A typical converged 2D URANS solution with approximately 0.1 million mesh elements is obtained in roughly 30 minutes on a quad-core work-station desktop computer. The converged 3D URANS solution with approximately 10 million mesh elements is obtained in roughly 54 hours on a quad-core work-station desktop computer.

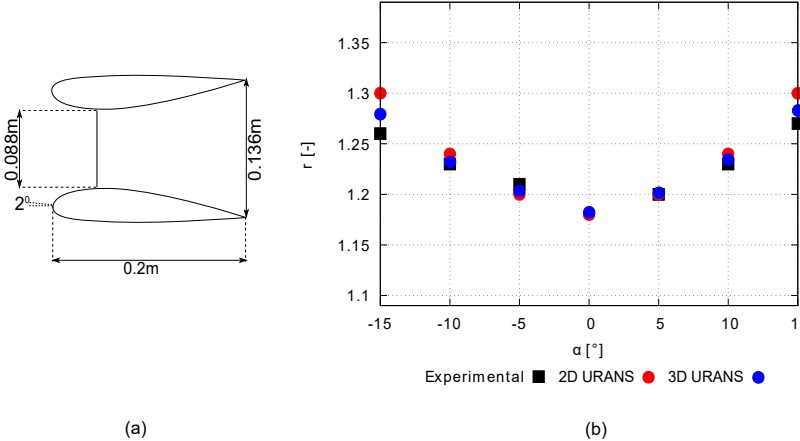

(a)                      (b)

**Figure 4.** A schematic cross-section layout of the three dimensional experimental model used for the numerical validation study (a), and comparison between experimental findings (Igra, 1981) and the CFD results (b).

## 4 Numerical verification and validation

For validating the numerical approach, experiments carried out by Igra (1981) on a duct-AD geometry (3-dimensional) are simulated. Igra's experiments were conducted in the subsonic wind tunnel of the Israel Aerospace Industry (formerly Israel Aircraft Industry); this tunnel has a large test section and it measures 3.6 m × 2.6 m.

A schematic of the cross-section geometry (named as Model B) is shown in Figure 4(a). The longitudinal cross-section of the duct is a NACA 4412 airfoil. The leading edge of the duct is rotated by 2° with respect to the free-stream direction, resulting in a duct expansion ratio (area of duct exit/area of the AD) of 1.54. A uniformly loaded AD model with $C_{T_{AD}} = 0.434$ is used to represent the turbine; the value is based on the selection of the author for the experiments. The experimental data set consists of: static pressure distribution at different axial and radial positions, and forces generated by the duct surface for a range of flow angles. During the experiments, the inflow velocity was set at $U_\infty$ = 32 m/s. Following Igra (1981), the wall interference and blockage correction can be ignored. The experimental data is reported in terms of the augmentation factor $r = \frac{C_P}{C_{P_o}}$, which expresses the ratio between the power coefficient of the duct-AD model and the power coefficient of the bare AD model when both the models bear the same AD and similar operating conditions.

A good agreement between the CFD simulations and the experimental findings is found in Figure 4(b). The deviation between the CFD and the experimental findings increase with increasing values of $\alpha$, especially for 2D URANS calculations.The differences in the 2D and 3D CFD results can be explained by looking at the flow field obtained using 3D URANS simulations. Figure 5 shows the time-averaged velocity contours of non-dimensional axial velocity $\frac{U_x}{U_\infty}$ in the $y - z$ plane at the AD location for Model B in non-yawed (left) and yawed (right) inflow conditions. Time averaging is performed over the quasi-steady solutions after convergence is reached. Because of the yaw angle ($\alpha = 10°$), an asymmetric flow field is present, thus the velocity at the AD plane changes with the azimuthal angle $\Phi$. Here, the azimuthal angle $\Phi$ is defined as positive in the clockwise direction when looking from upwind, with zero when oriented in the positive $y$ direction, see Figure 5 (left). The

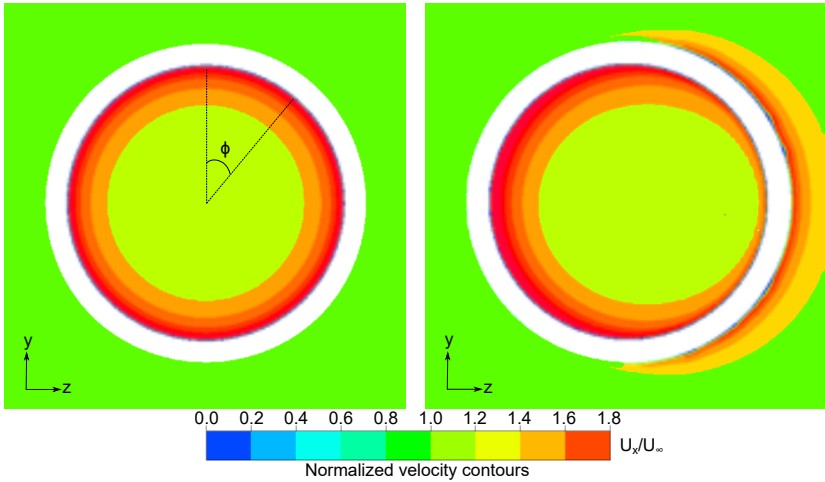

**Figure 5.** Contours of time-averaged non-dimensional free-stream velocity $U_x/U_\infty$ measured at the AD location located in the $y - z$ plane for Model B in (left) non-yawed inflow and (right) yawed inflow, $\alpha = 10°$.

main difference between the two results is due to fact that the $C_P$ (equation 6) obtained from 3D URANS simulations uses the azimuthally averaged streamwise velocity component, while the results from 2D simulations do not account for the gradual variation with $\Phi$. However, as shown in the comparsion, the three dimensional azimuthal effects are negligible when comparing $r$. It is important to highlight that the maximum deviation between 2D URANS results and experimental findings is less than

5% for $\alpha = \pm 15°$.

For an additional validation of the AD approach, numerical results obtained using 2D and 3D URANS are compared with the experimental study reported by Ten Hoopen (2009). The study was conducted using the full scale DonQi® DWT model in non-yawed inflow condition (see Figure 6). Experiments were conducted in the closed-loop open-jet (OJF) wind tunnel facility at the Delft University of Technology. The average thrust coefficient of the turbine $C_{T_{turbine}}$ was measured in the

experimental study to be 0.689; this value is chosen to model $C_{T_{AD}}$ for the results presented. Figure 6 shows the comparison of the normalized free-stream velocity $\frac{U_x}{U_\infty}$ measured behind the turbine blade at $x/c = 0.37$ in the radial direction $y$. Transition was not forced but the experimental model has a noise damper, see Figure 6, which acts as rough surface that forces transition to turbulence; this has not been replicated numerically. The computed velocity profiles preserves the overall shape, with the relative difference, calculated lower than 10%, which is within the experimental uncertainty and also attributed to the absence

of discrete blades and their related effects such as tip vortices, wake rotation and an accelerated mixing of the flow through the DWT with the external flow. An additional numerical verification exercise of the 2D URANS approach is performed, where the results are compared to a full-scale DWT numerical model. It is not reported herein for the sake of brevity; please refer to Appendix B

The 2D URANS approach gives results of reasonable accuracy when compared to the 3D URANS approach. The computing

cost issued by going from 2D URANS to 3D URANS does not justify the scope of the current study, where the effects of

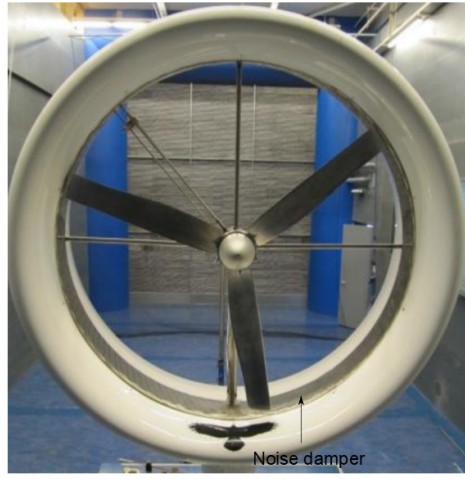
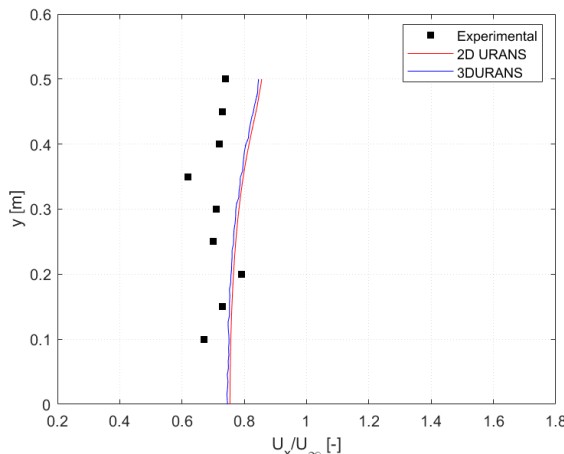

**Figure 6.** Comparison of dimensionless velocity profile vs radius (at x/c = 0.23) from center-line between the experimental data and the CFD findings shown for DonQi®DWT model in non-yawed inflow condition.

**Table 1.** Grid statistics for grid independence study of the reference case.

| Grid | Number of cells | $C_{T_D}$ |
|---|---|---|
| Coarse | 67640 | 0.3012 |
| Medium | 102008 | 0.3133 |
| Fine | 161028 | 0.3135 |

distributed AD loading, wake rotation and divergence are totally ignored. Having said that, the 2D URANS approach combined with numerical duct-AD model has been adopted for the results presented, hereinafter.

A grid independence analysis has been carried out for the 2D grid using three grid sizes, where the refinement factor in each direction is 1.5. Refinement factor is defined as the rate at which the grid size increases in the direction normal to the surface of the wall (duct surface). The duct thrust force coefficient $C_{T_D}$ is taken as reference for the convergence analysis. The results of the grid independence study are shown in Table 1. Convergence is reached for the medium refined grid, where the $C_{T_D}$ value fluctuates less than 0.0003%, and similar grid refinement is used in the numerical investigation, hereinafter.

## 5 Results and Discussion

### 5.1 Duct geometries

Two duct geometries, shown in Figure 7, with different longitudinal cross section (named as DonQi® and DonQi D5) are chosen for the current investigation. The selection is based on the duct shape parametrization study conducted by the authors (Dighe et al., 2019b). The parametrization procedure for duct shapes preserved the following geometric features: leading edge position

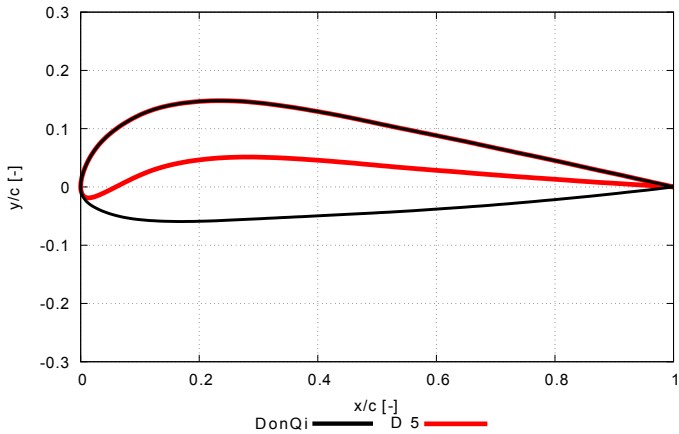

**Figure 7.** Cross-sectional geometry of the lower duct used for the numerical study.

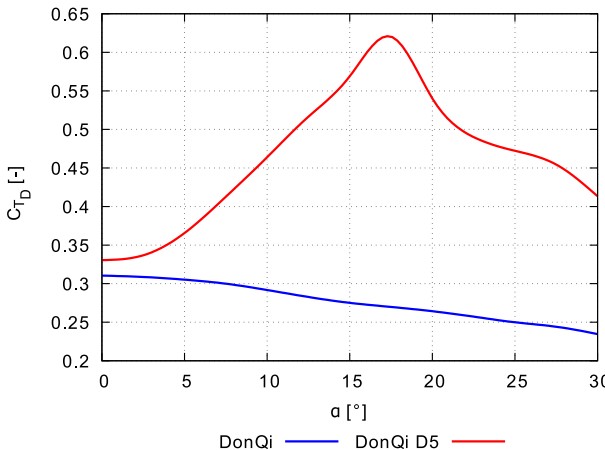

**Figure 8.** Effect of yawed inflow on the duct thrust force coefficient for the two duct geometries. $C_{T,AD} = 0.7$.

(which defines the inlet area ratio), trailing edge position (which defines the exit area ratio) and inner side thickness (which preserves AD radius and clearance). This makes it ideal to isolate the effects of the duct cross-section on the aerodynamic performance of the duct-AD model in yaw. In the study, an optimal $C_{T_{AD}} = 0.7$ was obtained for both the duct geometries. This value is employed for the rest of the discussion.

## 5.2 Duct force coefficient

Figure 8 illustrates the variation of duct force coefficient $C_{T_D}$ as a function of yaw angle $\alpha$ obtained for the two duct geometries investigated in this study. Starting with the $C_{T_D}$ trend-line for DonQi® duct, it can be observed that, $C_{T_D}$ decreases with increasing values of $\alpha$. Conversely, for DonQi D5 duct, $C_{T_D}$ increases with increasing $\alpha$. A local $C_{T_D}$ maximum at $\alpha = 17.5°$ appears for the DonQi D5 duct. The value of $C_{T_D}$ for DonQi D5 duct decreases for $\alpha$ beyond the local maximum.

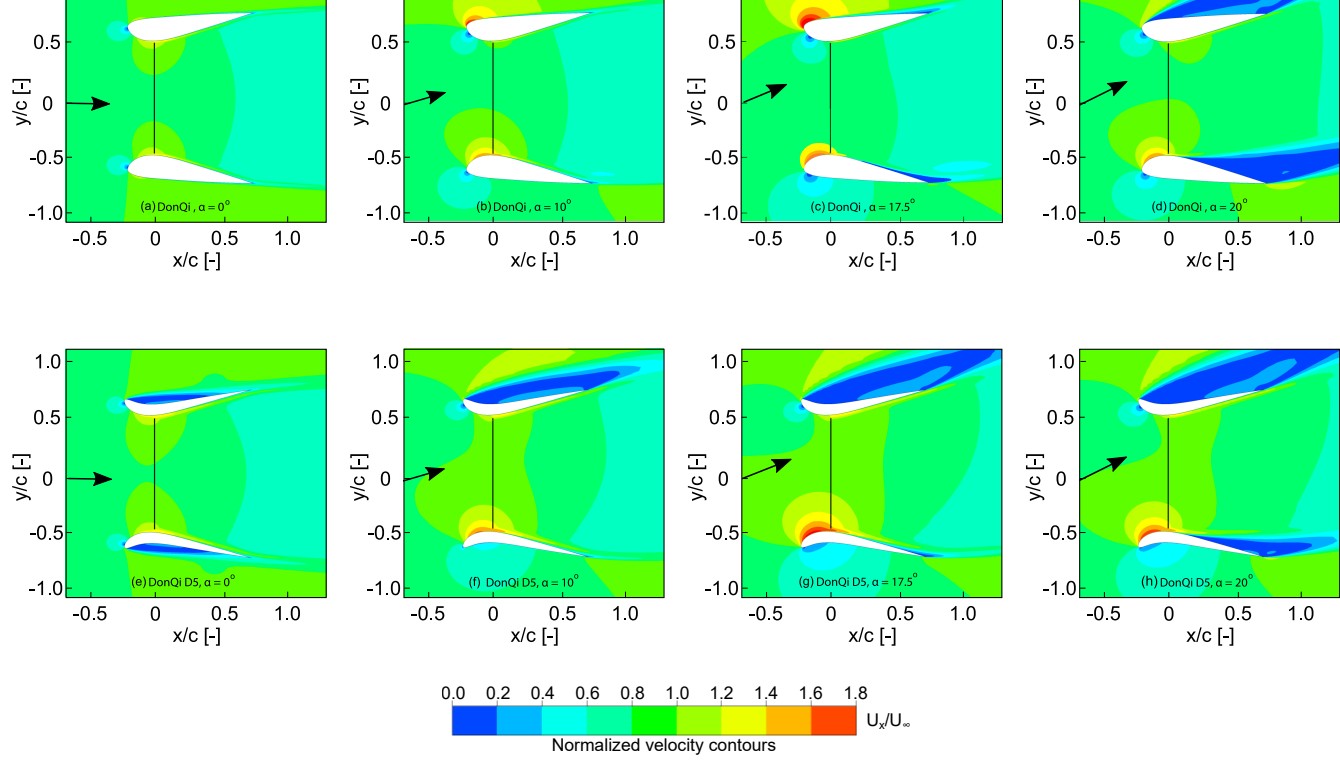

**Figure 9.** Velocity contours colored with streamwise normalized velocity. The results are depicted for DonQi duct-AD model (top) and DonQi D5 duct-AD model (bottom), both bearing a constant $C_{T,AD} = 0.7$.

The differences in the $C_{T_D}$ trend-lines for the two duct geometries can be explained by looking at the flow field. Contours of non-dimensional free-stream velocity $\frac{U_x}{U_\infty}$ for both duct geometries are reported in Figures 9 (a) to (h). A range of yaw angles have been tested, however, four yaw angles, i.e. $\alpha = 0°$, $10°$, $17.5°$ and $20°$, are presented here for the sake of conciseness. For the DonQi® duct configuration, the low pressure area, characterized by increased velocity, remain persistent inside and outside of the duct surfaces upto and including $\alpha = 17.5°$. The low pressure area, when seen outside of the duct surfaces, contribute negatively to the integrated duct thrust. For the DonQi D5 duct configuration, however, the low pressure area is limited on the inside of the duct surfaces, and high pressure area (characterized by reduced velocity) appear on the outside of the duct surfaces. The high pressure area is the result of the duct profile camber and is accompanied by flow separation, which adds positively to the duct thrust (see Figure fig:cx). At $\alpha = 20°$, where both DonQi® and DonQi D5 configurations are completely stalled, the resultant $C_{T_D}$ is higher for DonQi D5 duct. This is because the impact of stalled flow on the pressure side of the windward airfoil for DonQi D5 is larger since the stagnation pressure acts on the concave duct surface in comparison to the DonQi® duct surface, which is more convex. Hence, the resultant $C_{T_D}$ for DonQi D5 duct is much higher when compared with the DonQi® duct (see Figure 8) even though the general flow pattern in Figure 9 ($\alpha = 20°$) looks quite similar.

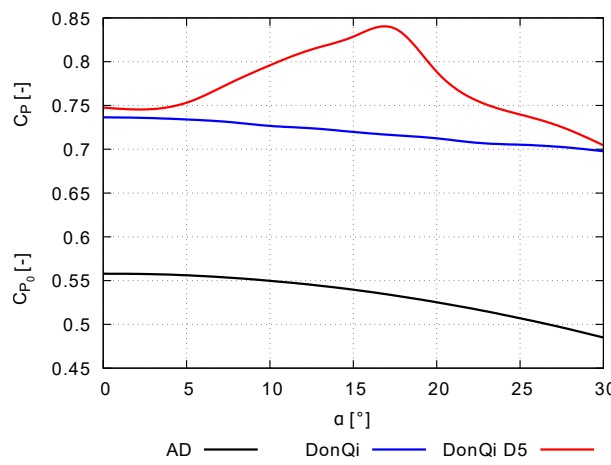

**Figure 10.** Effect of yawed inflow on the power coefficient.

### 5.3 Power Coefficient

Figure 10 represents the power coefficient $C_P$, for the two duct configurations, as a function of yaw angle $\alpha$. For the sake of completeness, $C_{P_o}$ for a bare AD is plotted alongside. The figure shows that, $C_P$ is higher than $C_{P_o}$ for all values of $\alpha$. Comparing Figures 8 and 10, the $C_P$ trends corresponds with the $C_{T_D}$ trends. The larger the $C_{T_D}$, the higher the $C_P$ reached,

and vice-versa. Similar to the $C_{T_D}$ trend for DonQi D5, maximum $C_P \approx 0.84$ is obtained for the DonQi D5 duct at $\alpha =$ 17.5°; thereafter any further increase in $\alpha$ results in $C_P$ drop. This also explains the experimental observations from Igra (1981), where a drop in the power coefficient for the duct-AD models with large duct expansion ratio was observed. For high duct expansion ratio, the likelihood of flow to separate from the inner walls of the duct increases (Abe and Ohya, 2004), thus lowering the $C_{T_D}$ and $C_P$ values for a given duct-AD model.

**6   Conclusions**

The present article reignites the insights of Igra (1981), Foreman and Gilbert (1983) and Phillips et al. (2002) to study the effects of yaw on the performance of DWTs. To this aim, two-dimensional numerical calculations using URANS simulations are performed. Based on the existing studies conducted by Dighe et al. (2019b, 2020), two duct geometries with different cross-section camber (named as DonQi® and DonQi D5) are chosen. To validate the numerical approach, comparison of the numerical

results with the experimental data are reported. Of the two duct geometries investigated, DonQi D5 duct configuration returns a gain in $C_P$ up to and including yaw angle $\alpha = $ 17.5°; thereafter any further increase in $\alpha$ results in the $C_P$ drop. On the contrary, $C_P$ of DonQi® duct configuration drops for $\alpha > 0°$. Flow field analysis pointed out that the aerodynamic performance of DWTs in yawed flow depend on the distinct shape of the duct under consideration. The high duct profile camber acts as a flow conditioning device and delays duct wall flow separation inside of the duct for a broad range of yaw angles. This phenomenon

is characterized by a rapid increase of duct thrust force coefficient $C_{T_D}$ and ultimately the $C_P$, for DonQi D5 configuration

in yaw. For the investigation presented here, a constant AD loading $C_{T_{AD}} = 0.7$ is chosen based on the optimization study presented in Dighe et al. (2019b). Future studies can investigate the effects of yaw on the performance of DWTs for a range of $C_{T_{AD}}$ values.

**Appendix A: Domain blockage study.**

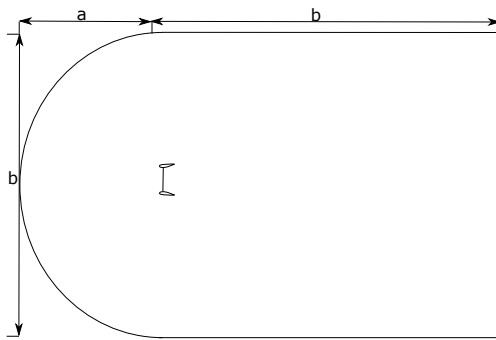

**Figure A1.** Schematic to describe the variables of the computational domain.

A major underlying factor that influences the accuracy and computational expense of CFD simulations is the size of the computational domain. For our current investigation, the size of the computational domain is defined by two variables; a and b (see Figure A1); where a is the upstream domain length from the AD location, and b is the total height of the domain and also the downstream domain length from the AD location; both the variables are normalized by the duct chord length c. The study is performed using the baseline DonQi® duct-AD model at yaw angle of 15°.

**Table A1.** Computational domain blockage study.

| Domain | a/c | b/c | $C_{T_D}$ |
|--------|-----|-----|-----------|
| 1 | 6 | 12 | 0.2748 |
| 2 | 12 | 24 | 0.2720 |
| 3 | 24 | 48 | 0.2719 |

The effect of computational domain sizes on the numerical prediction of $C_{T_D}$ is shown in Table A1. The unsteady simulations collects $C_{T_D}$ data, which is oscillating in time; time-averaged values obtained at quasi-steady-state are shown here. The $C_{T_D}$ values for domains 2 and 3 are almost identical, representing nearly unconfined conditions. The negligible difference can be attributed to the iterative convergence error or the computer round-off error. Domain 2 is chosen for the cases presented in this article.

**Appendix B: Numerical verification of the duct-AD model.**

Three-dimensional Lattice-Boltzmann Very Large Eddy Simulations (LB-VLES) of DWTs, where the rotor is simulated, in axial and yawed inflow conditions, forms the reference for the verification of the numerical approach presented in this article. For a detailed description of the LB-VLES approach, computational setup and operating conditions, the reader can refer to Dighe et al. (2020). The baseline DonQi® DWT model is simulated for $\alpha = 0°$ and $7.5°$. The free-stream velocity is $U_\infty =$

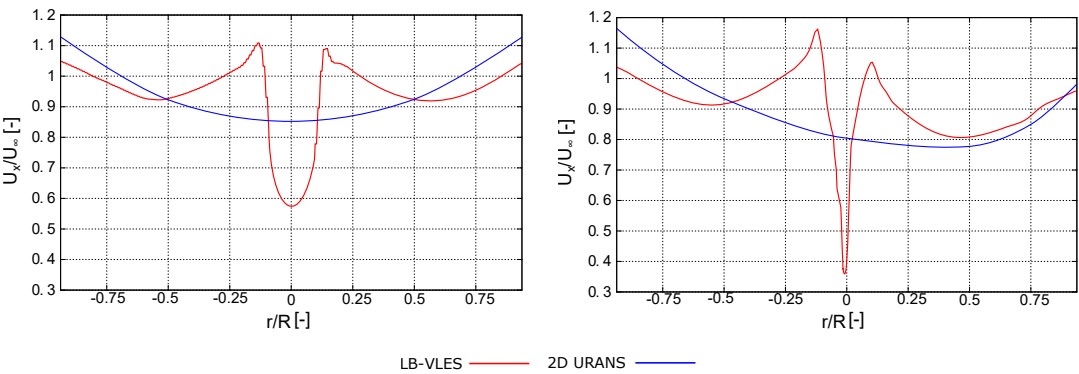

**Figure B1.** Radial distribution of streamwise velocity $U_x/U_\infty$ located just aft of the turbine/AD plane.

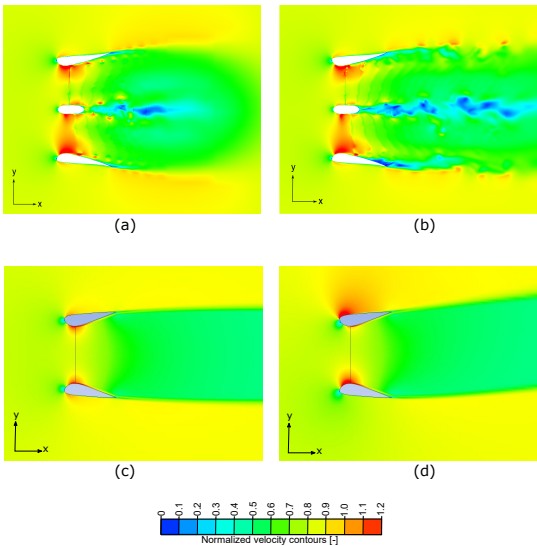

**Figure B2.** Contours of instantaneous streamwise velocity $U_x/U_\infty$ in the $x-y$ plane for (a) DonQi® at $\alpha = 0$ degrees using LB-VLES approach, (b) DonQi® at $\alpha = 7.5$ degrees using LB-VLES approach, (c) DonQi® at $\alpha = 0$ degrees using 2D URANS approach and (d) DonQi® at $\alpha = 7.5$ degrees using 2D URANS approach.

5 m/s, which corresponds to Reynolds number $Re = 3.31 \times 10^5$. Based on a previous study by Avallone et al. (2020), the resulting average rotor thrust coefficient equals 0.8; this value is adopted for specifying the input for the AD model.

Figure B1 examines the streamwise velocity component as a function of radial position using the two numerical approaches, both under non-yawed and yawed flow conditions. Before beginning this discussion, it must the stressed that the LB-VLES approach consists of turbine blades that are connected to a hub (upstream) and a nacelle (downstream). This geometric feature is not included in the duct-AD model; see Figure B2 (c) and (d). Despite this source of uncertainty in the AD modelling approach, viz. absence of discrete blade (including hub and nacelle) effects and wake rotation, the overall computed $\frac{U_x}{U_\infty}$ trends show good agreement. As a testimony to model skewed wake, as seen in Figure B2 (d), the 2D URANS duct-AD approach exhibits a strong potential to implicitly model the flow around a DWT in yaw. The proposed simplified approach thus captures first order flow physics; for higher order effects, the blade shape resolving models will be well suited.

*Author contributions.* VVD compiled the literature review, setup the CFD simulations and wrote the bulk of the paper. DS performed the CFD simulations, post-processed the cases and contributed towards writing this paper. FA reviewed the paper and carried out modifications in different sections of this paper. GVB helped formulate the ideas in regular group discussions.

*Competing interests.* The authors declare that they have no conflict of interest.

5  *Acknowledgements.* Authors would like to acknowledge Prof. Ozer Igra for providing the experimental data that have contributed to the part of numerical validation reported in this paper. The research is supported by STW organization, grant number- 12728

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
