# Peer review of "Ducted wind turbines in yawed flow: A numerical study"

_Wind Energy Science, 2019_

## Referee Comment (RC1) · Anonymous Referee #1 · 7 Oct 2019

Performance of the two different duct shape has been investigated for different yaw conditions. A verification study has been carried out. The performance of cambered duct shape is more than the non cambered duct shape with different yaw angles.

Its a good work.

———————————————

---

## Author Comment (AC1) · 9 Oct 2019

I, on behalf of my co-authors, would like to thank you for your time in reviewing the article.

————————————————————

---

## Referee Comment (RC2) · Anonymous Referee #2 · 18 Oct 2019

The aim of the considered paper is to check numerically the aerodynamic performance of ducted wind turbines in yawed flow conditions. As this was checked experimentally a long time ago, the present paper brings no new results. Should the used numerical simulation prove to be reliable, it could be a useful tool in designing future ducted wind turbines. In Fig. 5 of the considered submission experimental results taken from Igra (1981) are shown; it is apparent from this figure that the ducted wind turbine performance increases with increase in the duct's yaw angle, up to $\pm150$. It is also apparent from Fig. 5 that the duct's cross-section used in Igra's experiments looks like the presently simulated DonQi model. Surprisingly, in the results shown in Fig. 6 of the considered submission the DonQi duct experiences a decline in its performance (decline in CT,D) with increasing yaw angle. Is this a result of using a two-dimensional

flow model rather than the proper three-dimensional flow which is the appropriate case at yaw conditions? The explanations for the CT,D decline observed in model DonQi, given in Sec. 5.2, are based on results obtained using a two-dimensional flow model and therefore are physically questionable. Until the authors either conduct a three-dimensional flow solution or convince the readers that a two-dimensional flow model can correctly simulates the flow through the considered duct at yaw angles the present results are questionable. Based on the above comments, I leave the decision whether or not to accept the considered paper for publication in the Wind Energy Science Journal in the editors' hands.

---

## Author Comment (AC2) · 22 Oct 2019

The authors appreciate the efforts of the reviewer for the valuable comments. The paper has been modified following the reviewer suggestions. Modifications are reported in red in the revised manuscript.

1. As this was checked experimentally a long time ago, the present paper brings no new results.

The authors understand the comments of the reviewer. For this reason, the scope of the work has been rewritten to be more clear. The introduction has been expanded to highlight contrasting conclusions found in the literature that justify this study. The most important one is the missing physical explanation behind the different findings reported

in the literature.

2. Surprisingly, in the results shown in Fig. 6 of the considered submission the DonQi duct experiences a decline in its performance.

The results are consistent with other findings in the literature. For example, looking at the paper of Igra, six out of eight geometries showed insensitivity to yaw, whereas the other two showed a drop in the aerodynamic performance with the yaw angle; the performance drop was observed for the duct-AD models with high duct expansion ratio (>4.5). This point was not very clear in the first submission and has been clarified in the revised manuscript.

3. Until the authors either conduct a three- dimensional flow solution or convince the readers that a two-dimensional flow model can correctly simulates the flow through the considered duct at yaw angles the present results are questionable.

The authors understand the reviewer's concern. For this reason, 3D URANS simulations are added in the numerical validation section when comparing with the results of Igra. Although 3D URANS simulations increase the level of flow description by taking into account the three dimensional azimuthal effects, the computing cost increases by 4 times while the differences between 2D and 3D results are negligible. This is now clearly shown in the revised manuscript.

Please also note the supplement to this comment:
https://www.wind-energ-sci-discuss.net/wes-2019-62/wes-2019-62-AC2-supplement.pdf

---

## Referee Report (RR1)

Reviewer: M. Paul van der Laan, DTU Wind Energy

The authors perform 2D Unsteady Reynolds-averaged Navier-Stokes (URANS) simulation of a ducted Actuator Disk (AD) in yawed configurations for two different duct shapes. The 2D simulations are verified with a grid refinement study and with 3D simulations.

Please note that I am invited as a new reviewer, and I have not participated in the previous review rounds. Therefore, I have reviewed the latest revision as a first review, but I have also looked through the earlier reviews.

It is good that you have added a comparison between 2D and 3D simulations. Although, I am surprised to see that they agree so well for the yawed cases, since the flow problem is not symmetric, as pointed out by a previous reviewer. It is unclear to me how your AD forces are modelled in the 3D simulations. I guess you have used a uniformly loaded AD without tangential forces to make the comparison with the 2D simulations more fair? I expect that a more realistic thrust load distribution and tangential forces could have a significant effect on the results and I recommend to investigate this in the present work.

I have listed main and minor comments below, which should be addressed in order to accept the article as a publication for Wind Energy Science.

**Main comments**

1. What is the reason for performing URANS simulations? Couldn't you perform (steady-state) RANS simulations and obtain the averaged quantities directly? Or is the ducted AD an unsteady flow problem because of airfoil (that describes the cross section of the duct) is operating in stall for large yaw angles?

2. You mention that you model an AD with a uniformly distributed thrust load in the 2D simulations. Using a uniform loaded AD is often fine for wind farm simulations, where one is interested in (far) wake interactions. However, the current work focuses on the aerodynamics of a ducted AD and then I would recommend a more realistic AD model, especially when you investigate the effect of yaw misalignment on the forces. Have you investigated more realistic loading distributions? For 3D simulations, you could use an AD based on airfoil data. In case you lack airfoil data, you could also use an AD based on a Joukowsky rotor as proposed by Sørensen et al. (2020), which performs similarly as an AD based on airfoil data for conventional wind turbines (at least without a diffuser). Both AD loading models also provide the power directly by integrating the tangential forces instead of using an estimate (eq. 6).

3. It is unclear to me how your AD forces are modelled in the 3D simulations. Do you also assume a constant thrust load and no tangential forces? If yes, could this have been the reason why the 2D and 3D simulations compare well?

4. I was wondering if there exist literature where a ducted wind turbine has been simulated by resolving the geometry of both the duct and wind turbine in the numerical grid. If one had such a model available, then one could verify the loading of the present setup where the wind turbine is modeled as an AD.

5. Page 3, Line 16: You mentioned that you model the AD with an infinitesimal width, which makes sense for a 2D simulation, but not for 3D simulation. Please clarify in the text.

6. The domain size is not very large or at least much smaller than typical 2D airfoil simulations that my colleagues perform using DTU's inhouse CFD code EllipSys and an o-grid with a radius of 40 to 50 chord lengths. Have you investigated

the effect of this relatively small domain on the results ($C_T$ and $C_P$)? A small domain could induce additional blockage effects that could lead to an increased mass flow through your AD.

7. You could split the methodology section in to two subsections, describing the 2D and 3D CFD setups, separately, which could clarify some of my previous questions.

8. What are the Reynolds numbers of the all validation cases and how do they compare with the typical Reynolds number of utility scale urban wind turbines?

9. Page 4, Line 10: You mention that the Betz limit of $C_P$ can be exceeded for a ducted wind turbine, which is indeed true. You could refer to the work of Sørensen (2016), where analytic formulations of $C_P$ are discussed in Section 3.4 (in case you can access this reference).

10. You could add 3D CFD results in Figures 9 and 10, since they are you most important figures that support your main conclusions.

**Minor comments**

1. Line 21: This is → There are.

2. You could rephrase the title of Section 4 as *Numerical verification and validation*, since you both verify the model (using a grid refinement study and a 3D setup) and validate the model with wind tunnel measurements.

**References**

Sørensen, J.: General Momentum Theory for Horizontal Axis Wind Turbines, Research Topics in Wind Energy, Springer, 2016.

Sørensen, J. N., Nilsson, K., Ivanell, S., Asmuth, H., and Mikkelsen, R. F.: Analytical body forces in numerical actuator disc model of wind turbines, Renewable Energy, 147, 2259, https://doi.org/https://doi.org/10.1016/j.renene.2019.09.134, 2020.

---

## Referee Report (RR2)

**Review of *Ducted wind turbines in yawed flow: A numerical study* by Vinit Dighe et al. (R5)**

Reviewer: M. Paul van der Laan, DTU Wind Energy

Thanks the for corrections. I have two suggestions for the newly added content.

**Minor comments**

1. Page 8, Line 25: I believe you should refer to Appendix B (not Appendix A). It seems that you forgot to refer to the blockage study of Appendix A in the text.

2. Appendix A: It is great that you have added results of the center line velocity for two different domain sizes. It would probably be more relevant to report the effect of the domain size on $C_{T_D}$, similar to the grid refinement study. In addition, I would recommend to use at least three domain sizes, each with a factor two enlargement.

---

## Author Response (AR2)

The authors appreciate the valuable comments from reviewers #3 and #4. The manuscript has been modified following the reviewers comments. Modifications are reported in blue in the revised manuscript. For the sake of completeness, modifications carried following the comments of reviewers #1 and #2 in the first round of peer-review are retained and highlighted in red.

1. Response to reviewer #3:

   - **Page 1, line 22: correct the sentence (there are ...)**
     The sentence is corrected.

[Figure]

- **Page 3, line 2: why not including rotating actuator disc? and why not 3D?**
  Through this study, the authors wished to characterize the performance of the DWT in yawed flow without accounting for 3D effects. A 2D AD approach allows us to decouple the effects discrete blades from the turbine, while also saving a singnificant amount of computational resources. Thus, the effects of distributed AD loading, wake rotation and divergence are totally ignored for this preliminary investigation. The study would lay a foundation for more detailed analysis of DWTs in yaw, where 3D and rotational effects can be included.

  Following your comments, the authors performed a further numerical validation of the numerical approach presented. Numerical results are compared with the experiments on a full 3D DWT model (see page 7, line 13). The computed results agree well with the experimental findings, with the relative difference, calculated lower than 10%, which is within the experimental uncertainty.

- **Page 4, 18: you mention "minimum" y+ of 1: do you mean "maximum" here? If your simulations are wall resolved, your yplus should be less than 1. Can you plot the y+ values across the ducts for the investigated cases? Same line: you mention standard wall function is used. Why do you use wall function at all? if your mesh is fine enough to resolve the boundary layer, wall functions should not be used.**
  Indeed, this has not stated transparently in the previous version of the paper. The maximum value of yplus along the duct walls is 1. Standard wall functions have not been used and this has been removed from the current version of the paper. The distribution of $y+$ on the duct wall surface is shown in Figure 4.

- **Page 4, line 21: Please explain what is a "fan" boundary condition. It is not a classic BC.**

[Figure]

Details about the fan boundary condition is added (see page 5, line 4).

- **Page 5, line 8: "multi-core" is a vague explanation. Explain how many cores is exactly used for the computations. Same page: line 10: add "camber" after cross-section. Can you explain what does D5 mean here?**
  The corrections are made (see page 6, line 5) and (page 9, line 3).

- **Page 6, fig 5: Please add both upper and lower sections of the duct (and/or write in the caption that this is the lower cross-section) and possibly with the use of colors make it more clear which lines correspond to which duct. It is a little unclear until you see figure 9.**
  Figure 7 has been modified.

- **Page 6, line 15: why can the blockage effect be ignored in this scenario?**
  Numerical results are validated against the experiment reported by Igra (1981). Following the explanation by Igra, the experimental test section nozzle is large (10 times that of the experimental model used), and therefore does not suffer from interference or blockage effects.

- **Page 7: provide more details on the 3D case, a snapshot of the mesh, lateral extent, etc.**
  Figure 3 is modified in order to include the lateral extent of the 3D computational domain

- **Page 7, line 5: Provide more details on time averaging (how many iterations used for it? are figure 9 the time-averaged velocities or the instantaneous ones?)**
  Further details are added (see page 6, line 2). The velocities are time-averaged.

- **Page 8, line 7: four times larger, but how many mesh points?**
  A more clear explanation is provided (see page 6, line 5).

[Figure]

- **Page 10, line 14: Ok for the explanation of the fact, but can we hear what is the physics behind?**
  Further details are added (Page 11, line 16).

- **Page 11, line 6: This sentence looks a little paradoxical to me: "... DonQi D5 duct configuration not only demonstrates an insensitivity to yaw but a gain in the overall performance" does it alter the performance or is it insensitive after all?**

  The sentence has been corrected.

2. Response to reviewer #4:

- **The paper attempts to study the impact of yawed wind on ducted wind turbines. This study is fundamentally flawed because the real problem is three-dimensional (and not even axisymmetric). Therefore, 2D simulations have no relevance (2D axisymmetric assumptions may be OK for non-yawed conditions, but these are not even axisymmetric). The fact that 3D and "2D" simulations agree for the validation case is just coincidence. Unless detailed 3D studies are conducted, this paper must be rejected.**

  The study is an attempt to understand the aerodynamic performance of DWTs in yaw for different duct configurations. For the current investigation, the effects of distributed AD loading, wake rotation and divergence are totally ignored. The study would lay a foundation for more detailed analysis of DWTs in yaw, where 3D and rotational effects are included (for e.g. Dighe, V. V., Avallone, F., van Bussel, G. (2020). Effects of yawed inflow on the aerodynamic and aeroacoustic performance of ducted wind turbines. Journal of Wind Engineering and Industrial Aerodynamics, 201, 104174.). The

agreement between the 2D and 3D results suggests that the simplified duct-AD approach to model the flow around DWTs in yaw was satisfactory, and should be considered in the preliminary stages. Following your comments, changes have been made to the manuscript:

- Additional references using the AD approach to study yawed inflow for simple HAWTs have been added (see page 3, line 3).
- A further numerical validation of the numerical approach is presented. Numerical results are compared with the experiments on a full 3D DWT model (see page 7, line 13). The computed results agree well with the experimental findings, with the relative difference, calculated lower than 10%, which is within the experimental uncertainty.
* * *

---

## Editor Decision (ED2)

Wes 2019-62

Overall

Abstract
- It is not self-evident that yawed flow conditions must be characterized – elaborate
- Can you provide quantitative values regarding the results in terms of performance
- Can you add a one sentence explanation for WHY the increase and subsequent drop ocurrs?

Introduction
- Enforced visual and noise regulations are not the main reasons why we don't see broad uptake of small / urban wind. LCOE for small wind in urban settings is not competitive with large utility-scale wind farms for several reasons. Some are the environmental and operational conditions for small wind (as you note), others are economies of scale, technology learning and more
- Why DWTs over non DWTs for small urban wind? It is not well justified. A duct can introduce speed ups but the duct itself presents a large additional capital cost. A sentence or two more in motivation would be helpful
- How prevalent are situations operating in yawed flow conditions for small wind DWTs? I would think that passive or active yaw systems would largely obviate this operational condition except for infrequent situations… what does the standard IEC 61400-2 saw about this?
- What is the point of the bare turbine left figure in Figure 1? It is not helpful. Better would be to show two images of DWT, one in normal inflow and one in yawed inflow conditions
- It is strange that in Gilbert and Foreman they saw now change in performance with yaw angle up to 30 degree. This does not agree with recent results in the literature from a myriad of sources. See literature from Paul Fleming, Pieter Gebraad, Jennifer King, Jan-Willem van Wingerden, and many more… you should reference the literature on operation with yaw offset for normal turbines since it is quite relevant and also extensive and recent
- Use of URANS is insufficiently motivated.  What did Phillips and company use?

Duct – AD flow model
- Consider making this a subset of section 3 on methodology and computational setup

Methodology and Computationaal setup
- Might also point out why not using LES for the simulations. They are more expensive but when you are investigating physical phenomena, it is often best to start with highest fidelity – i.e. is URANS enough? Explain why and also explain the limitations of using URANS instead of LES

Numerical verification and validation

- Explanation of using 2D instead of 3D URANS for analysis is still week even after updates. Figure 6 shows decent divergence of the experimental and simulation data – comparison with other fidelity analysis tools (i.e. time-averaged stats of LES) would be helpful. Appendix A moves in this direction. I don't think this should be appendix.
- It is important to the overall work. it would be better to elaborate on appendix A statement "despite this source of uncertainty, the overall…." In what ways good agreement? And what are the reasons for the lack of agreement?

Results and discussion
- There are issues in figure 9 – what is going on there? It makes it impossible to read this section
- Cp goes up with the yawed conditions, it would be good to discuss in conclusions / future work about potential impacts on loading
- Something is missing here in terms of discussing the novelty of the findings in the context of other work – why do we care about these results?

Conclusions
- These need to be strengthened considerably – do not use bullets. Speak more critically of the work in the context of the study limitations and also tie into a discussion on future work.

---

## Author Response (AR3)

The authors appreciate the valuable comments from reviewers 5 and 6. The manuscript has been modified following the reviewer's comments. Modifications are reported in green in the revised manuscript. For the sake of completeness, modifications carried following the comments of reviewers 1 and 2 in the first round of peer-review are retained and highlighted in red, and modifications carried following the comments of reviewers 3 and 4 in the second round of peer-review are retained and highlighted in blue.

Response to reviewer 5

- What is the reason for performing URANS simulations? Couldn't you perform (steady-state) RANS simulations and obtain the averaged quantities directly?
  Large flow separation regions are expected for DWTs in yawed flow depending on the geometry of the duct and the yaw angle. Flow solutions obtained using steady RANS formulation for DWTs with large yaw angles did not converge or even diverge. Using URANS formulation, the goal was to capture the asymptotic behavior (quasi-steady-state) of the flow in order to reach a converged solution. The selected approach also takes the learning from Apsley Leschziner (2000), who investigated the ability of unsteady simulations using the $k - \omega$ SST model to predict separated flows in a duct and compared them to experimental data. The agreement between the experimental and computation results was found to be good, and therefore the approach was chosen for the current investigation.

- I would recommend a more realistic AD model, especially when you investigate the effect of yaw misalignment on the forces. Have you investigated more realistic loading distributions?
  In all the simulations presented in this article, the turbine is represented using a numerical actuator disc (AD) model, a method widely used to model the principal effects of the turbine in a simplified manner. In a recent study by the authors (see: Dighe, Vinit V., Francesco Avallone, and Gerard van Bussel. "Effects of yawed in-flow on the aerodynamic and aeroacoustic performance of ducted wind turbines." Journal of Wind Engineering and Industrial Aerodynamics 201 (2020): 104174.), it was shown that the azimuthal variation of axial velocity at the rotor radial plane was relatively low. Incorporating the azimuthal effects using more sophisticated techniques like actuator line or actuator surface methods would definitely enable more accurate calculations of the local induced velocities. However, one should note that the flow physics for a bare turbine and a turbine within a duct is completely different. Incorporating the real rotor geometry/distributed loading for modeling turbine effects during this preliminary investigation would not allow us

to decouple the turbine and the duct effects, thus preventing a proper analysis of the yawed flow for DWTs. To this aim, the simplified AD approach is chosen deliberately for this investigation, so as to study the impact of duct shapes and not the specific performance of a rotor within a duct.

- Do you also assume a constant thrust load and no tangential forces?
  A uniformly (constant) loaded AD model without tangential loading is used to represent the turbine. The main difference between the 2D and the 3D results is that the 3D URANS simulations use the azimuthally averaged streamwise velocity component, while the results from 2D simulations do not account for the gradual variation with the azimuthal angle.

- I was wondering if there exists literature where a ducted wind turbine has been simulated by resolving the geometry of both the duct and wind turbine in the numerical grid. If one had such a model available, then one could verify the loading of the present setup where the wind turbine is modeled as an AD.
  The current article is meant to be a preliminary investigation to study the aerodynamics of DWTs in yawed flow, and particularly on the effect of the duct geometry on the aerodynamic performances. A comparison study with the real DWT model (a subsequent paper published by the authors), in which both the duct and the turbine have been simulated will certainly improve the quality of the numerical verification and validation. Having said that, the numerical verification exercise is added in the article as an appendix.

- You mentioned that you model the AD with an infinitesimal width, which makes sense for a 2D simulation, but not for 3D simulation. Please clarify in the text.
  This has been rectified.

- Have you investigated the effect of this relatively small domain on the results?
  A detailed study to investigate the blockage effects due to the varying domain size has not been investigated in the current research. However, in a previously

published research by the authors (Dighe, V. V., Avallone, F., Igra, O., Bussel, G. V. (2019). Multi-element ducts for ducted wind turbines: a numerical study. Wind Energy Science, 4(3), 439-449.), a similar numerical setup was validated with experiments where the wall interference and blockage correction can be ignored. A good agreement was seen.

- You could split the methodology section into two subsections, describing the 2D and 3D CFD setups, separately
  The section has been revised for clarity.

- What are the Reynolds numbers of all validation cases and how do they compare with the typical Reynolds number of utility-scale urban wind turbines?
  In the context of existing commercial DWT models, the Reynolds number range from 200,000 to 1,000,000 depending on the model geometry. The current numerical study is performed at a fixed $Re$ of $4.5 \times 10^5$.

---

## Author Response (AR4)

The authors appreciate the valuable comments from the reviewers. The manuscript has been modified following the reviewer's comments. Modifications are reported in orange in the revised manuscript. For the sake of completeness, modifications carried following the comments of reviewers 1 and 2 in the first round of peer-review are retained and highlighted in red, modifications carried following the comments of reviewers 3 and 4 in the second round of peer-review are retained and highlighted in blue and modifications carried following the comments of reviewers 5 and 6 in the third round of peer-review are retained and highlighted in green.

1. it is not possible to refer to an experimental work to verify your numerical domain

size. You have to verify this by performing simulations with different domain sizes and report the influence on Ct and Cp. If you are not able to perform such a study then you need to state in the article that the relative small domain might have influenced the results and future work is necessary to quantify it.

To address this remark, a study on domain blockage effects is now included in Appendix A.

2. The Reynolds-numbers that you have mentioned to me cannot be found in the revised version, please add them, because this is important information for those who would like to redo your simulations.

The Reynolds number is now explicitly stated in the article; i.e. $Re$ of $4.5 \times 10^5$.

3. Has this been settled? The end of this paragraph seems to suggest that the wind-aligned flow through a ducted turbine is well understood, which this statement appears to contradict. Also: What about blade tip aerodynamics and induction effects? Are the diffusers designed to ensure flow attachment throughout the near wake? Please elaborate.

To address these remarks, Section 1 (Introduction) is modified and the modifications are highlighted in orange.

4. I believe "tau" in the caption should be "Gamma." Also, what you've drawn is the circulation in the wake, which is equal and opposite to the actual bound circulation.

Figure 1 is modified.

5. This equation does not appear to have consistent units–the LHS is dimensionless while the RHS has units of velocity.

Equation 2 is corrected.

6. This is unclear to me. $C_{T_D}$, which is the force exerted by the duct on the flow, is equal and opposite to the drag force exerted by the wind on the duct. Per the discussion in

the introduction, the increased mass flow should arise from the lift generated by the duct.

To this aim, the related text is re-written in order to avoid confusion.

7. Are you running DES? Otherwise, this is overkill.

The authors appreciate this valuable remark. However, the chosen y+ value is based on the User manual provided by Ansys (see https://www.afs.enea.it/project/neptunius/docs/fluent/html/ug/node410.htm)

8. Is this assumption valid, especially given yawed inflow (e.g., as you've sketched out in Fig 2) at > 20 deg, as discussed later in Section 5?

The AD approach is chosen deliberately for this fundamental research/investigation, so as to study the impact of duct shapes, and not the specific performance of a rotor within a duct. In order to assess the validity of the AD approach, an additional numerical verification exercise of the 2D AD approach is performed in Appendix B, where the results are compared to a full-scale DWT numerical model. The results show that the simplified AD approach is suitable to capture the first order flow physics; for higher order effects, the blade shape resolving models will be well suited.

9. Where does this come from? It appears quite low.

The value is based on the cited author's choice for the experiments, and is used for numerical validation in our study.

10. Is Cp calculated by multiplying the rotor-averaged velocity with the input Ct? Did you run each case with and without the diffuser geometry?

$C_P$ reported here is based on the equation 6, and $\frac{U_{AD}}{U_\infty}$ is the integral of the velocity across the AD surface. Yes, for calculating $r$ (augmentation factor), simulations for AD without duct are performed but the numbers not reported here explicitly for the sake of brevity.

11. Unclear what value this additional exercise adds. It shows that the wake deflection as well as the flow around the far diffuser wall (at +y) differs significantly between the 2D URANS and 3D VLES.

The following exercise was based on the recommendation of a reviewer in order to assess the validity of the AD approach.

12. At what distance downstream of the rotor? Are the 2D and 3D results similarly comparable further downstream?

The downstream distance is now indicated in Figure 6. Unfortunately, no additional measurements were available further downstream to extend our numerical validation.

13. How does the duct thrust continuously decrease for the DonQi design? For example, comparing alpha=20 in Fig 9, there is massive flow separation on both the top and lower parts of the duct; I would expect the thrust to be >= the thrust on the D5 at this point.

The authors highly appreciate for addressing this detail. At $\alpha = 20°$, where both DonQi® and DonQi D5 configurations are completely stalled, the resultant $C_{T_D}$ is higher for DonQi D5 duct. This is because the impact of stalled flow on the pressure side of the windward airfoil for DonQi D5 is larger since the stagnation pressure acts on the concave duct surface in comparison to the DonQi® duct surface, which is more convex. Hence, the resultant $C_{T_D}$ for DonQi D5 duct is much higher when compared with the DonQi® duct even though the general flow pattern ($\alpha = 20°$) looks quite similar. A more detailed explanation is added as modifications in the manuscript.

---

## Author Response (AR5)

**Comments**

August 11, 2021

The authors appreciate the valuable comments from Dr. Paul van der Laan. Modifications are carried out in the revised manuscript.

- I believe you should refer to Appendix B (not Appendix A). It seems that you forgot to refer to the blockage study of Appendix A in the text.
  Both appendixes (A and B) are now correctly stated in the main article.

- It is great that you have added results of the center line velocity for two different domain sizes. It would probably be more relevant to report the effect of the domain size on $C_{T_D}$, similar to the grid refinement study. In addition, I would recommend to use at least three domain sizes, each with a factor two enlargement.

  Following the recommendation, Appendix A is revised.

---

## Author Response (AR6)

**Response to Associate Editor**

August 27, 2021

The authors appreciate the valuable comments from the Associate Editor (AE) in order to improve the quality of the work. Modifications are carried out in the revised manuscript, which are highlighted using purple text.

- Abstract

    - AE: It is not self-evident that yawed flow conditions must be characterized – elaborate. Can you provide quantitative values regarding the results in terms of performance. Can you add a one sentence explanation for WHY the increase and subsequent drop ocurrs?

    - Abstract: The abstract is now modified in order to provide a quantitative value for the performance rise with a concise explanation of the aerodynamic phenomenon studied.

- Introduction:

    - AE: Enforced visual and noise regulations are not the main reasons why we don't see broad uptake of small / urban wind. LCOE for small wind in urban settings is not competitive with large utility-scale wind farms for several reasons. Some are the environmental and operational conditions for small wind (as you note), others are economies of scale, technology learning and more. Why DWTs over non DWTs for small urban wind? It is not well justified. A duct can introduce speed ups but the duct itself presents a large additional capital cost. A sentence or two more in motivation would be helpful. How prevalent are situations operating in yawed flow conditions for small wind DWTs? I would think that passive or active yaw systems would largely obviate this operational condition except for infrequent situations. . . what does the standard IEC 61400-2 saw about this?

    - The AE indicated the importance of cost to be highlighted in the ducted wind turbine (DWT) design space. To this aim, a recent study (Design considerations for a small DWT) providing a detailed investigation on the parameters related to the cost of the DWT system is now included in the reference list. In our current manuscript, the discussion is now limited to the parameters relating to the (aerodynamic) performance.

- AE: What is the point of the bare turbine left figure in Figure 1? It is not helpful. Better would be to show two images of DWT, one in normal inflow and one in yawed inflow conditions.

- Figure 1 includes a schematic of flow around a bare wind turbine and a DWT in order to highlight the importance of the additional duct thrust force, which remains dominant in our discussion. The figure aids the reader to draw a parallel comparison between the performance of a bare and ducted system; performance of bare system in yaw is also indicated in Figure 10 of this manuscript for the sake of completeness.

- AE: It is strange that in Gilbert and Foreman they saw now change in performance with yaw angle up to 30 degree. This does not agree with recent results in the literature from a myriad of sources. See literature from Paul Fleming, Pieter Gebraad, Jennifer King, Jan-Willem van Wingerden, and many more. . . you should reference the literature on operation with yaw offset for normal turbines since it is quite relevant and also extensive and recent.

- To our best knowledge, we have included all studies highlighting the effects of yaw on the performance of DWT. In order to avoid confusion to the readers, the discussion did not include the references studying the yaw offsets in relation to bare wind turbine performance.

- AE: Use of URANS is insufficiently motivated. What did Phillips and company use?.

- Following AE's comment, the motivation for the use of the URANS approach is highlighted. Also, the limitations of the numerical study performed by Phillips et al. is now included.

- Duct – AD flow model:

  - AE: Consider making this a subset of section 3 on methodology and computational setup.

  - The authors believe that section 2, which deduces the aerodynamic performance parameters for DWT, be a separate section. Section 3 details the numerical methodology and the numerical setup. Including section 2 within section 3 would create interruption in the flow for the readers.

- Methodology and Computational setup:

  - AE: Might also point out why not using LES for the simulations. They are more expensive but when you are investigating physical phenomena, it is often best to start with highest fidelity – i.e. is URANS enough? Explain why and also explain the limitations of using URANS instead of LES.

- – Following AE's comment, we now start the discussion with LES simulations performed in this context. Then, the choice of URANS approach for our current investigation is identified.

- Numerical verification and validation:

  - – AE: Explanation of using 2D instead of 3D URANS for analysis is still week even after updates. Figure 6 shows decent divergence of the experimental and simulation data – comparison with other fidelity analysis tools (i.e. time-averaged stats of LES) would be helpful. Appendix A moves in this direction. I don't think this should be appendix.

  - – For our main discussion, we try to limit our numerical validation study of the URANS approach with two independent wind tunnel experiments. The additional validation in Appendix B was performed to highlight the flow characteristics, in particular the skewed wake, being captured using URANS approach and LES simulations.

  - – AE: It is important to the overall work. it would be better to elaborate on appendix A statement "despite this source of uncertainty, the overall. . . ." In what ways good agreement? And what are the reasons for the lack of agreement?

  - – We now explicitly highlight the uncertainties involved in the numerical validation study (Appendix B) showing comparison between a duct-AD model (URANS approach) and a DWT model (LES approach).

- Results and discussion:

  - – AE: There are issues in figure 9 – what is going on there? It makes it impossible to read this section. Cp goes up with the yawed conditions, it would be good to discuss in conclusions / future work about potential impacts on loading. Something is missing here in terms of discussing the novelty of the findings in the context of other work – why do we care about these results?

  - – The text in this section is further modified in order to offer clarity to the readers.

- Conclusions:

  - – AE: These need to be strengthened considerably – do not use bullets. Speak more critically of the work in the context of the study limitations and also tie into a discussion on future work.

  - – Following AE's comments, this section has been completely re-written, which now includes discussion on the future work.